# Transfer Learning Approaches for Brain Metastases Screenings

**DOI:** 10.3390/biomedicines12112561

**Published:** 2024-11-08

**Authors:** Minh Sao Khue Luu, Bair N. Tuchinov, Victor Suvorov, Roman M. Kenzhin, Evgeniya V. Amelina, Andrey Yu. Letyagin

**Affiliations:** 1The Artificial Intelligence Research Center of Novosibirsk State University, Novosibirsk State University, 630090 Novosibirsk, Russia; 2FSBI Federal Neurosurgical Center, 630087 Novosibirsk, Russia; 3Research Institute of Clinical and Experimental Lymphology, Branch of the Institute of Cytology and Genetics, Siberian Branch of Russian Academy of Sciences, 630060 Novosibirsk, Russia; 4Center for New Functional Materials, Novosibirsk State University, 630090 Novosibirsk, Russia

**Keywords:** transfer learning, brain metastases, segmentation

## Abstract

Background: In this study, we examined the effectiveness of transfer learning in improving automatic segmentation of brain metastases on magnetic resonance imaging scans, with potential applications in preventive exams and remote diagnostics. Methods: We trained three deep learning models on a public dataset from the ASNR-MICCAI Brain Metastasis Challenge 2024, fine-tuned them on a small private dataset, and compared their performance to models trained from scratch. Results: Results showed that models using transfer learning performed better than scratch-trained models, though the improvement was not statistically substantial. The custom Tversky and Binary Cross-Entropy loss function helped manage class imbalance and reduce false negatives, limiting missed tumor regions. Medical experts noted that, while fine-tuned models worked well with larger, well-defined tumors, they struggled with tiny, scattered tumors in complex cases. Conclusions: This study highlights the potential of transfer learning and tailored loss functions in medical imaging, while also pointing out the models’ limitations in detecting very small tumors in challenging cases.

## 1. Introduction

Brain metastasis (BM) happens when cancer cells spread to the brain, creating secondary tumors that interfere with normal brain function [1,2]. These tumors usually indicate an advanced stage and high malignancy of cancer, making treatment more difficult and lowering the chances of long-term survival. BM is serious because even small brain tumors can be deadly by pressing inside the skull or causing serious problems if they are in critical areas [3]. Survival rates are generally low, especially without early detection and intervention. Even with treatment, the prognosis can vary depending on tumor characteristics (number and type of tumors), patient factors (age), and treatment options (surgery, radiation, and chemotherapy) [4].

Diagnosis and treatment of BMs often require advanced imaging techniques like magnetic resonance imaging (MRI) to identify the size, location, and number of tumors [5]. MRI is especially useful because it provides clear and detailed images with excellent soft tissue contrast, allowing doctors to detect small tumors, assess swelling around them, identify perinodular bleeding within the brain, and understand how they affect important parts of the brain. Moreover, MRI does not use ionizing radiation, making it safer for patients who need multiple scans [6]. Common MRI sequences, such as T1-weighted (T1W), post-contrast T1-weighted (T1C), and T2-weighted (T2W) scans, reveal the brain’s structure, while Fluid-Attenuated Inversion Recovery (FLAIR) highlights areas of swelling and lesions near cerebrospinal fluid spaces. Advanced MRI techniques like diffusion-weighted imaging and perfusion imaging offer additional insights into tumor characteristics, such as cellularity and blood flow [6]. By using these sequences, MRI helps distinguish between healthy and abnormal tissues, guiding early personalized diagnosis and effective treatment planning for brain metastases. Using AI to automatically measure and track BM volumes following SRS treatment, this study showed a strong correlation between AI-driven measurements and the current clinically used method: manual axial diameter measurements [7].

Precise segmentation of BMs helps doctors accurately assess tumor size, location, and spread, which is essential in improving the results of surgery and radiation therapy [8]. By clearly defining tumor boundaries, treatments can be targeted more effectively while reducing harm to healthy brain tissue. Segmentation also allows doctors to track tumor changes over time, helping to evaluate how well treatments are working. However, manual segmentation is time-consuming, prone to errors, and can vary between doctors, leading to inconsistent results.

Deep learning (DL) [9] has made significant strides in medical image analysis by enabling automated segmentation of brain MRI [10,11,12,13,14]. A recent review demonstrated the role of machine learning and deep learning methods in lesion detection, diagnosis, and anatomical segmentation of various types of brain tumors, including metastases [15]. Building on these advancements, several novel approaches have been proposed to further improve the accuracy and reliability of brain metastasis segmentation. Grøvik et al. used a fully convolutional neural network based on a 2.5D GoogLeNet architecture to automatically detect and segment brain metastases from multisequence MRI, achieving an overall AUC of 0.98 and a Dice score of 0.79, with an average false-positive rate of 8.3 per patient [16]. Huang et al. proposed the DeepMedic+ network with a custom volume-level sensitivity-specificity loss, which significantly improved brain metastasis detection by increasing sensitivity from 0.853 to 0.975 and precision from 0.691 to 0.987, while reducing false positives by 0.444 [17]. Highlighting the importance of strategic modality selection and multi-stage processing, Sadegheih et al. [18] proposed a two-stage detection and segmentation model using T1C, T1W, and FLAIR modalities, which significantly improved brain metastasis segmentation accuracy compared to single-pass models. Yang et al. trained the 3D-TransUNet model for brain metastases segmentation, exploring both Encoder-only and Decoder-only configurations with Transformer self-attention, and found that the Encoder-only version with Masked-Autoencoder pre-training achieved a lesion-wise Dice score of 59.8% [19]. In addition to these model-based improvements, new tools have been developed to bring advanced segmentation methods into clinical practice. An open-source software, Raidionics (v1.2.3), was also developed to perform preoperative segmentation of major brain tumor types, allowing standardized clinical reports to be generated in about ten minutes. It achieved an average Dice of 0.85 with 0.95 recall and precision for preoperative segmentation, while postoperative performance was lower, with an average Dice of 0.41 [20].

Despite these advances, DL faces challenges, particularly the limited availability of high-quality annotated medical images. Transfer learning (TL) [21] can overcome these limitations by allowing models trained on large datasets to be adapted for specific tasks with less training data, based on the idea that many basic features are shared across different image types. This includes various techniques like instance-based, network-based, and adversarial-based transfer [22]. TL has been applied to a wide range of medical imaging tasks—such as segmentation, object identification, disease classification, and severity grading [23].

Several recent studies have introduced advanced deep learning techniques for brain tumor segmentation, using pre-trained models and improved architectures to boost performance. Wacker et al. enhanced the AlbuNet architecture for brain tumor segmentation using a 3D U-Net-based model with a ResNet34 encoder, pre-trained on ImageNet, and extended convolutional layers to process volumetric MRI data. The 3D model with pretraining achieved higher Dice scores and more stable training than the 2D model, but its performance was less consistent on a new clinical dataset due to data differences [24]. Tataei Sarshar et al. introduced a deep learning pipeline for brain tumor segmentation using a pre-trained ResNet50 model with a multi-modality approach, combined with cascade feature extraction, inception, and new mReLU blocks to enhance learning. The proposed method achieved mean Dice scores of 0.9211 for the tumor core, 0.8993 for the whole tumor, and 0.9223 for the enhancing region [25].

Further extending these innovations, a recent study by Messaoudi et al. [26] proposed embedding pre-trained 2D networks into higher-dimensional U-Nets for effective segmentation of 2D and 3D medical images, utilizing weight and dimensional transfer techniques, and demonstrated superior performance in benchmarks. Huang et al. presented a federated learning framework using Learning Without Forgetting to train deep learning models for brain metastasis segmentation across multiple medical centers without sharing raw data, achieving strong performance on diverse MRI datasets [27]. Pani and Chawla introduced a hybrid approach that integrates transfer learning and self-supervised learning within a 3D UNET architecture to improve brain tumor segmentation in MRI scans, effectively reducing the need for extensive annotated data while achieving high accuracy with a Dice score of 90.15 [28].

While deep learning models have shown promise in automatically segmenting brain metastases, their application in clinical settings is often limited by a lack of alignment with radiologist expertise. Most existing studies evaluate model performance based solely on quantitative metrics but do not consider feedback from medical experts, who are key to judging how useful and accurate the results are in practice. Our research addresses this by including radiologists’ and physicians’ feedback in evaluating fine-tuned models, ensuring that the results are both accurate and practical for clinical use.

## 2. Materials and Methods

### 2.1. Overall Process

We started by training three deep learning models on a public dataset of brain metastases MRI, using 5-fold cross-validation. The private dataset with a smaller number of images was divided into a training set (26 images) and a testing set (10 images). We fine-tuned the selected models on the private training set using pre-trained weights on the public dataset and also trained them from scratch on the private data to compare performance. Both the fine-tuned and scratch-trained models were tested on 10 unseen samples. We employed explainable techniques to visualize the segmentations and gathered detailed feedback from medical experts on the results, focusing on their clinical relevance and accuracy. This helped us assess how well the models performed in segmenting brain metastases after fine-tuning compared to training from scratch. Figure 1 depicts our overall process.

### 2.2. Data Sources and Preparation

In this study, we employed two distinct datasets. The first dataset, ASNR-MICCAI Brain Metastasis Challenge 2024 [29], included MRI scans of untreated brain metastases collected from multiple institutions under standard clinical conditions. This dataset provided four MRI sequences: T1W, T1C, T2W, and FLAIR. Initial segmentations were generated by automated algorithms, then manually refined by neuroradiologists, and finally approved by board-certified specialists. The dataset used a three-label segmentation system, classifying regions as non-enhancing tumor core (NETC), surrounding FLAIR hyperintensity (SNFH), and enhancing tumor (ET). The challenge provided a training dataset with annotations and a validation dataset without annotations. In this study, we selected 652 metastasis cases from the training dataset, which we referred to as the BraTS dataset.

The second dataset, the Siberian Brain Tumors (SBT), was sourced from the Federal Neurosurgical Center in Novosibirsk, Russia. This dataset included clinical data and MRI scans (T1W, T1C, T2W, FLAIR) from 496 patients, covering a range of diagnoses such as glioblastomas, astrocytomas, neurinomas, meningiomas, and metastases. For this study, only the metastasis cases (36 in total) were utilized. MRI scans were predominantly acquired using a 1.5T Siemens MAGNETOM Avanto MRI machine (manufactured by Siemens Healthineers, Erlangen, Germany), with some cases imaged at 3T Philips Ingenia MRI system (manufactured by Philips Healthcare, Amsterdam, Netherlands). Annotations for this dataset included peritumoral edema, non-enhancing tumor regions, GD-enhancing tumor regions, and necrotic tumor cores. These annotations were manually conducted by two board-certified neuroradiologists following a strict protocol, with an independent expert reviewing and correcting any errors. Ground truth labels were cross verified with patient clinical histories, as well as histological and immunohistochemical data. All cases were formatted in NIFTI, with 192 slices of 512 × 432 pixels, co-registered to a standard template, and resampled to a 1 × 0.5 × 0.5 mm^3^ resolution. The SBT data were divided into 26 training cases and 10 test cases.

The key feature is the availability of information about the source of the oncological process and medical imaging of metastases in the brain. During the anonymized preprocessing stage, unique IDs were assigned to each test sample, and these IDs were maintained across all experiments to facilitate verification and enable restoration of the original data when necessary. The test set IDs are 726, 727, 728, 729, 730, 731, 732, 733, 734, and 739.

In this study, the terms “case” and “sample” were used interchangeably, though with slight nuance: “case” referred to the overall patient, specifically a single patient diagnosed with brain metastasis, while “sample” denoted an individual instance within the dataset associated with that patient. Overall, when referring to a “case” or “sample”, it did not indicate a single MRI scan but rather the complete set of four MRI sequences that collectively represented one patient with brain metastasis.

All models used the default nnU-Net preprocessing to ensure the input data were prepared consistently and effectively. This preprocessing included adaptive resampling, which adjusts depending on the dataset’s characteristics, and dynamic intensity normalization, which changes the normalization method based on data variability. Foreground cropping was improved to better focus on the important areas, reducing extra computation. Padding and patch size were made flexible, adjusting to the GPU memory for maximum efficiency. Adaptive patch spacing was used to optimize the resolution during training, balancing efficiency and detail. Lastly, dynamic data sampling helped manage class imbalance, especially useful for datasets with different tumor sizes.

### 2.3. Model Architecture and Configuration

We employed the nnU-Net framework [30] for all model implementations, leveraging its self-configuring U-Net architecture. nnU-Net was chosen because of its ability to adapt automatically to different datasets by optimizing key architectural parameters such as depth, number of layers, and patch sizes, which are crucial in medical image segmentation tasks where data variability is high. The encoder-decoder structure, along with skip connections, ensures that spatial information is preserved, which is critical for precise tumor boundary detection.

For this study, we employed three models: (1) the default nnU-Net configuration (Default), (2) nnU-Net with a custom combined loss function (TverskyBCE), and (3) SegResNet [31]. Three graphs depicting the architectures used, including the dimensions of each layer, activation functions, normalization techniques, and other architectural details, were provided in the Appendix A to offer a comprehensive understanding of the model configurations.

The Default model was used without modifications, dynamically adapting to the dataset by optimizing patch size, input modalities, and network depth. Its self-adjusting capability allowed it to achieve optimal performance across different datasets, justifying its use as a baseline. The initial learning rate was set to 1 × 10^−2^, and the optimizer employed was stochastic gradient descent (SGD) with Nesterov momentum (momentum = 0.99). A polynomial learning rate scheduler was applied to gradually reduce the learning rate as training progressed, ensuring smooth convergence and preventing overshooting.

The TverskyBCE configuration modified the nnU-Net architecture by employing a combined loss function that merges Tversky loss [32] with Binary Cross-Entropy (BCE) loss. Tversky loss, designed to handle class imbalance, prioritized false negative reduction—critical for detecting small lesions. The Tversky loss parameters were set at α = 0.3 and β = 0.7, emphasizing recall over precision. To further enhance sensitivity to small tumors, BCE loss was applied with a positive class weight (pos_w = 10), focusing more on correctly identifying tumor regions. The two losses were combined at a 1:1 ratio to optimize both small tumor detection and overall segmentation quality. The optimizer and learning rate scheduler used for the TverskyBCE model were identical to those of the Default model.

SegResNet was implemented with a U-Net-like architecture, enhanced by residual connections to improve gradient flow and prevent vanishing gradients, ensuring stability during the training of deeper networks. The encoder consisted of 4 stages, with residual blocks (1, 2, 2, 4) capturing progressively complex features, while the decoder used 1 residual block per stage for reconstruction. The model started with 32 filters, which increased in deeper layers to capture finer details. Input and output channels were configured based on the number of input modalities and segmentation classes. The Adam optimizer with a learning rate of 1 × 10^−4^ and weight decay of 1 × 10^−5^ was used to ensure efficient parameter updates. A polynomial learning rate scheduler was employed to gradually reduce the learning rate, preventing overfitting and ensuring smooth convergence.

### 2.4. Training and Fine-Tuning

All models were trained using GPU acceleration with the 3d_fullres configuration from the nnU-Net framework. The training spanned 200 epochs for pre-training and 100 epochs for fine-tuning. To enhance generalization and robustness, data augmentation included spatial transformations (such as rotations and scaling), intensity adjustments (including brightness, contrast, and gamma modifications), noise injections (e.g., Gaussian noise, Gaussian blur), and techniques like low-resolution simulation and mirroring.

For pre-training, models were trained on the BraTS dataset using training plans aligned with those of the SBT dataset, ensuring consistency and facilitating smoother transfer learning. During fine-tuning, models initialized with pre-trained weights from BraTS were further trained on the SBT dataset. Additionally, we trained models on the SBT dataset from scratch for 100 epochs to evaluate the benefits of fine-tuning compared to training from scratch.

All training was conducted on three workstations with GPUs (Quadro RTX 8000, NVIDIA GeForce GTX TITAN X, and NVIDIA RTX A4500), using Python 3.10.

### 2.5. Evaluation and Validation

To evaluate the performance of our segmentation models, we used several common metrics in medical image analysis, including Dice Similarity Coefficient (DSC), Hausdorff Distance (HD), Sensitivity (SEN), and Specificity (SPE) [33]. DSC measures overlap between predicted and ground-truth segmentations (0 to 1, higher is better). HD assesses boundary quality (lower is better). SEN and SPE measure the ability to correctly identify true positives and true negatives (both range from 0 to 1, higher is better). We also calculated true and predicted volumes to evaluate how well the model captures lesion size. For testing, we ensembled the predictions from all fold models and used a separate hold-out set of 10 images for an unbiased evaluation.

For the quantitative evaluation, we conducted a comparative analysis of the models using descriptive statistics, including the mean, standard deviation, and 95% confidence intervals. These statistical measures provided insights into the average performance, consistency, and reliability of the fine-tuned models relative to the scratch-trained models. To determine whether the data met the assumptions for parametric testing, we first applied the Shapiro–Wilk test [34] to assess whether the distribution of metrics for all models followed a normal distribution. Based on the results of the normality test, we observed that some of the metrics did not follow a normal distribution. As a result, we employed a one-sided Mann–Whitney U test [35], a non-parametric statistical test, to evaluate whether the performance scores of the fine-tuned models were stochastically greater than those of the scratch-trained models. The alternative hypothesis (H_1_) posited that the fine-tuned models would outperform the scratch-trained models in terms of evaluation metrics. We reject the H_1_ if the *p*-value from the Mann–Whitney U test is greater than or equal to the significance level of 0.05, indicating that there is insufficient evidence to support that the fine-tuned models outperform the scratch-trained models.

For the qualitative evaluation, we selected the two best models from our quantitative analysis and sent their predictions on 10 test cases to our medical experts (radiologists and physicians). The segmentation results were evaluated by medical experts using 3D Slicer v5.6.2 [36] software to visually assess the quality and accuracy of the predicted tumor boundaries. After their independent assessment, a follow-up meeting was held to discuss the models’ strengths and weaknesses, particularly in handling complex tumor characteristics. This discussion helped identify areas for improvement to enhance clinical use.

### 2.6. Data and Code Availability

The BraTS dataset presented in the study is openly available at https://www.synapse.org/Synapse:syn59059764 (accessed on 15 July 2024).

The SBT dataset presented in this article is not readily available because it contains private health data collected in our Federal Neurosurgical Center and sharing it publicly would violate patient privacy and confidentiality agreements in accordance with ethical and regulatory standards.

The code to reproduce this study is available at https://github.com/luumsk/BrainMetaSeg.git (accessed on 5 November 2024).

### 2.7. Manuscript Preparation

An AI-assisted tool, ChatGPT (powered by OpenAI’s GPT-4o model), was used to assist in refining the text of the manuscript. The tool was employed exclusively to improve language clarity and grammar, without altering the scientific content or interpretations provided by the authors.

## 3. Results

### 3.1. Quantitative Analysis of Model Performance

#### 3.1.1. Descriptive Statistics

Our analysis showed that fine-tuned models generally performed better than scratch-trained models in key areas such as segmentation accuracy, sensitivity, and boundary localization. Fine-tuned models achieved higher DSC, lower HD for enhancing the tumor and tumor core, and better SEN. They also demonstrated more consistent and reliable predictions, indicated by narrower confidence intervals and lower standard deviations. Although scratch-trained models had a slight advantage in specificity and boundary delineation for the whole tumor, fine-tuning proved to be more effective in the most important aspects of segmentation.

Specifically, fine-tuned models had higher DSC across all tumor regions: 0.905 compared to 0.902 for enhancing tumor, 0.936 compared to 0.934 for tumor core, and 0.914 compared to 0.907 for whole tumor. Figure 2 supports this trend, showing that fine-tuned models consistently achieved higher mean DSC across different configurations, with similar or narrower error bars, suggesting improved accuracy and stability. Although the differences were modest, the results suggest that fine-tuning offers clear benefits for segmentation accuracy and consistency, particularly for enhancing tumor and tumor core regions.

Figure 3 further shows that fine-tuned models generally provided better boundary localization than scratch-trained models, especially for enhancing tumor and tumor core. The fine-tuned model had a mean HD of 2.76 mm for enhancing tumor and 1.87 mm for tumor core, compared to 2.78 mm and 5.50 mm, respectively, for the scratch-trained model. For the whole tumor, scratch-trained models performed slightly better, with a mean HD of 3.23 mm versus 3.36 mm for fine-tuned models. Despite this, fine-tuned models showed lower variability for enhancing tumor and tumor core, indicating more consistent results.

Among the transfer learning approach with fine-tuned model variants, TverskyBCE demonstrated the best overall performance for brain metastasis segmentation. It achieved higher DSC and superior boundary localization compared to the Default model, showcasing a clear advantage in segmentation quality. While SegResNet also performed well, it lacked consistency, particularly in capturing fine boundary details. TverskyBCE offered the optimal balance of accuracy and stability, making it the preferred choice for enhancing segmentation quality in this context.

#### 3.1.2. Statistical Testing for Significance

The statistical test results indicated no significant difference between the fine-tuned and scratch-trained models across all metrics. However, it is important to note that the sample size in this study is relatively small, consisting of only 10 cases approved by the histological and immunohistochemical evaluations. This small sample size reduces the statistical power of the test, increasing the likelihood of failing to detect real performance differences that may exist between the models. Despite the lack of statistical significance, the fine-tuned models consistently showed slightly higher performance across all metrics, with improvements in DSC, lower HD, and better SEN and SPE. Also, all experiments were focused on evaluating the clinical significance and application of some explainable techniques. These slight improvements suggest that fine-tuning does contribute positively to segmentation performance, particularly in complex tasks such as brain tumor segmentation. In practical applications, where even small improvements in accuracy and boundary precision can be clinically meaningful, these results are still relevant and important meaning for the screening process.

To further compare model variants across sub-regions, we run the Mann–Whitney U test on pairs of models, with the alternative hypothesis that the scores of the first model were significantly greater than those of the second. The results indicated that the TverskyBCE model outperformed both SegResNet (*p* = 0.04) and Default (*p* = 0.03) in whole tumor segmentation. Additionally, TverskyBCE exhibited statistically significant higher SEN across all sub-regions compared to both SegResNet (*p* = 0.02) and Default (*p* = 0.01). These findings suggested that TverskyBCE offered superior performance in whole tumor segmentation and SEN metric.

### 3.2. Individual Case Analysis and Visual Comparison

For each model and metric, we identified the cases with the lowest DSC, SEN, SPE, and the highest HD, labeling them as challenging cases. To find the test samples that most models struggled with, we counted how often each sample was marked as a challenging case. Finally, we selected the top three most challenging cases (ID 731, 733, 739 from the test set SBT data) across all models for further visual analysis.

Test sample 733 was identified as the most challenging case, having been marked 28 times across models. The tumor is small, spans only a few slices, and is centrally located in the brain. Most models, including the TverskyBCE scratch-trained model, failed to correctly segment the tumor core. However, the TverskyBCE Fine-tuned model performed significantly better, as shown in Figure 4, achieving the highest DSC and the lowest HD for this case. The Tversky loss function helped the model focus on reducing false negatives, which is essential for capturing small tumor regions. However, the full potential of this loss function became evident only after fine-tuning. Fine-tuning allowed the model to further refine its weights, adapting more closely to the specific dataset and learning critical patterns necessary for detecting small, subtle lesions. The combination of the Tversky loss and fine-tuning enabled TverskyBCE Fine-tuned to successfully segment the tumor core, which was missed by the scratch-trained version and other models. This highlights the complementary roles of the Tversky loss function and fine-tuning in enhancing SEN to challenging cases.

Test sample 731 was identified as the second most challenging case, marked 11 times across models. The tumor is large, located on one side of the brain with significant surrounding non-enhancing FLAIR hyperintensity. While all models detected the tumor boundary well, as indicated by the low HD for both the tumor core and whole tumor, most models struggled with the enhancing tumor, missing the middle region, which resulted in low SEN. The TverskyBCE Fine-tuned model performed best, with the smallest amount of missing tumor tissue. Although the DSC and SEN scores did not fully reflect the visual difference, the TverskyBCE Fine-tuned model clearly showed fewer missing regions in the images, as shown in Figure 5. This demonstrates that the combination of Tversky loss and fine-tuning was particularly effective at reducing false negatives and capturing more of the enhancing tumor.

Test sample 739 was marked 11 times as a challenging case. The tumor is very large, covering a significant portion of the brain and spanning multiple slices. While most models segmented the boundary reasonably well for both the enhancing tumor and the tumor core, they struggled with the segmentation of the whole tumor, resulting in a high HD. The whole tumor’s irregular, heterogeneous shape, with jagged edges and asymmetrical extensions, as shown in Figure 6, added to the difficulty. The TverskyBCE Fine-tuned model performed best in this case, achieving the highest DSC and SEN, along with the lowest HD for the whole tumor.

The analysis showed that the TverskyBCE Fine-tuned model outperformed other models, particularly in challenging cases with small or irregular tumors. The combination of Tversky loss and fine-tuning was crucial in reducing false negatives and improving segmentation accuracy. Fine-tuning enabled the model to better adapt to the dataset, allowing it to capture complex tumor boundaries more effectively.

### 3.3. Expert Feedback and Clinical Relevance

After evaluating the predictions of the TverskyBCE Fine-tuned and SegResNet Fine-tuned models on 10 test cases, medical experts provided feedback based on medical visualization (evaluation) and comparisons with ground truth annotations. Overall, both models performed well. In cases with large, straightforward lesions, such as neuroendocrine metastases (test sample 730) and undifferentiated cancer metastases (test samples 734 and 739), there were minimal differences between the models’ results. However, both models struggled with recognizing additional small tumors, often missing tiny lesions in more complex cases. Metastasis is grown from cancerous cells which spread to the brain from the affected area by blood. The imbalanced objects and regions issue affects the data-driven learning algorithm as the extracted features may be highly influenced by large tumors and additional small parts. For example, to the regions—the necrotic/non-enhancing tumor core region is much smaller than other regions; to the objects—a characteristic feature of metastases is their multicentrality, they can appear in several areas of the brain with different consequences (size of lesions).

In test sample 728 (breast metastases), displayed in Figure 7, both models achieved high quantitative metrics, including a high DSC (0.90–0.92) and low HD (0.8–2.0). However, upon review by medical experts, both models failed to detect tiny tumors, which are critical for accurate diagnosis and treatment. Missing these small tumors is especially concerning in metastatic breast cancer, as even tiny lesions can indicate early-stage metastasis or tumor progression, which directly impacts treatment decisions and prognosis. This discrepancy highlights the limitations of relying solely on quantitative metrics and indicates that comparative metrics may not fully capture the models’ performance in detecting small, clinically important lesions.

In test sample 729 (adenocarcinoma metastases), TverskyBCE outperformed SegResNet in segmenting a cerebellar tumor. While SegResNet showed less edema in its segmentation, TverskyBCE’s result was closer to the ground truth, capturing the tumor’s characteristics more accurately. Similarly, in test sample 733 (low-grade cancer metastases), TverskyBCE successfully detected the necrotic tumor core, a critical sub-region that SegResNet entirely missed. Necrosis within a tumor is crucial for diagnosis and prognostic evaluation. This comparison is visualized in Figure 8, where the differences between the models are shown.

In summary, expert feedback indicated that while both models performed reliably in simpler cases, TverskyBCE demonstrated a clear advantage in handling more complex scenarios, particularly in recognizing the necrotic core and accurately managing intricate tumor features. These findings highlight the strengths of each model and point to areas for improvement, especially in detecting small tumors and complex tissue structures.

## 4. Discussion

In this study, we assessed the performance of transfer learning models for brain metastasis segmentation. We pre-trained nnU-Net (with default settings and with Tversky and BCE loss) and SegResNet on the BraTS Metastases 2024 dataset, and then fine-tuned these models on our private SBT dataset. We compared the fine-tuned models with those trained from scratch, using both quantitative and qualitative analyses, to obtain feedback from medical experts on the segmentations produced. This expert input was crucial for evaluating clinical accuracy and ensuring the models were fit for practical use.

Our main objective was to develop methodologies for diagnosing and screening metastases. Brain metastases occur when cancer cells spread to the brain from the affected area. Any cancer can spread to the brain, but lung cancer, breast cancer, colon cancer, kidney cancer, and melanoma are the most likely to cause brain metastases. Tumors that experts call high-grade gliomas are tumors of the central nervous system (CNS). They are solid tumors and appear due to a mutation of brain or spinal cord cells. Since these tumors grow in the central nervous system, they are also called primary CNS tumors. That is, they are not metastases from other malignant tumors that have grown in other organs, and their cancer cells have penetrated the central nervous system. An additional feature is that the lesions can be small—difficult to detect with the human eye. The diagnosis is highly dependent on the quality of the image, the thickness of the slices, the scanning parameters, and the expertise of the doctor.

Throughout this study, several key challenges were identified in the context of brain metastasis segmentation:-Location and morphological uncertainty: Metastases are grown from cancer cells which spread to the brain from the affected area by blood. Due to the wide spatial distribution of cancer cells, lung cancer, breast cancer, colon cancer, kidney cancer, and melanoma may appear at any location inside the brain. The shape and size of different brain tumors vary with large morphology uncertainty. Each sub-region of a meta may also vary in shape and size.-Low contrast: High resolution and high contrast images are expected to contain diverse image information. Due to the image projection and tomography process, MRI images may be of low quality and low contrast. The boundary between biological tissues tends to be blurred and hard to detect.-Annotation bias: Manual annotation highly depends on individual experience, which can introduce an annotation bias during data labeling. The annotation biases have a huge impact on the AI algorithm during the learning process.-Imbalanced region and object issues: The imbalanced issue affects the data-driven learning algorithm as the extracted features may be highly influenced by large tumor regions. For example, to the regions—the necrotic/non-enhancing tumor core (NCR/ECT) region is much smaller than another region; to the objects—a characteristic feature of metastases is their multicentrality, they can appear in several areas of the brain with different consequences (size of lesions).

Our results showed that fine-tuning the SBT dataset led to a slight improvement in metrics compared to training from scratch. This suggests that transfer learning is helpful when labeled data are limited, as pre-trained models use features learned from larger datasets. However, the performance differences were not statistically significant, likely due to the small test size (10 samples). Despite this, qualitative analysis showed that the fine-tuned models, especially TverskyBCE, performed better in challenging cases. TverskyBCE detected small or irregularly shaped tumors and did not miss the tumor core, unlike other models. This suggests that a customized loss function like Tversky with Binary Cross-Entropy helps handle class imbalance and improves sensitivity for smaller lesions. Medical experts also confirmed that TverskyBCE detected critical tumors in some cases where SegResNet did not.

However, all models, including TverskyBCE and SegResNet, struggled with detecting tiny tumors. Experts emphasized the need to improve the detection of smaller lesions and boundary accuracy, despite the models achieving good overall segmentation quality. Even with high quantitative metrics, both models failed to detect small tumors, revealing that metrics alone are insufficient to capture clinically important issues. The involvement of medical experts is crucial in identifying these gaps, as missing tiny tumors can significantly impact treatment decisions and prognosis. Our findings not only demonstrate the potential of transfer learning for brain metastasis segmentation but also highlight the importance of medical expert input in developing AI algorithms. Their insights help bridge the gap between quantitative performance and real-world clinical needs, ensuring AI models are refined to detect smaller tumors and improve boundary segmentation for more accurate treatment planning.

Looking ahead, future research could address these limitations by applying advanced post-processing techniques to improve boundary delineation and using synthetic data generation or data augmentation to enhance tiny tumor detection. Incorporating multi-modal imaging data, such as combining MRI with CT or PET, could also provide more comprehensive information and boost model performance. Additionally, exploring new model architectures or hybrids could further improve segmentation, particularly for small or complex lesions. Another important direction would be to consider including edema in segmentation, as brain edema often surrounds metastases and could provide a critical context for more accurate tumor delineation. Finally, ongoing collaboration with medical experts will be key to refining models and ensuring they meet clinical standards, with expert feedback incorporated into the model development process to continuously improve performance in real-world applications.

## Figures and Tables

**Figure 1 biomedicines-12-02561-f001:**
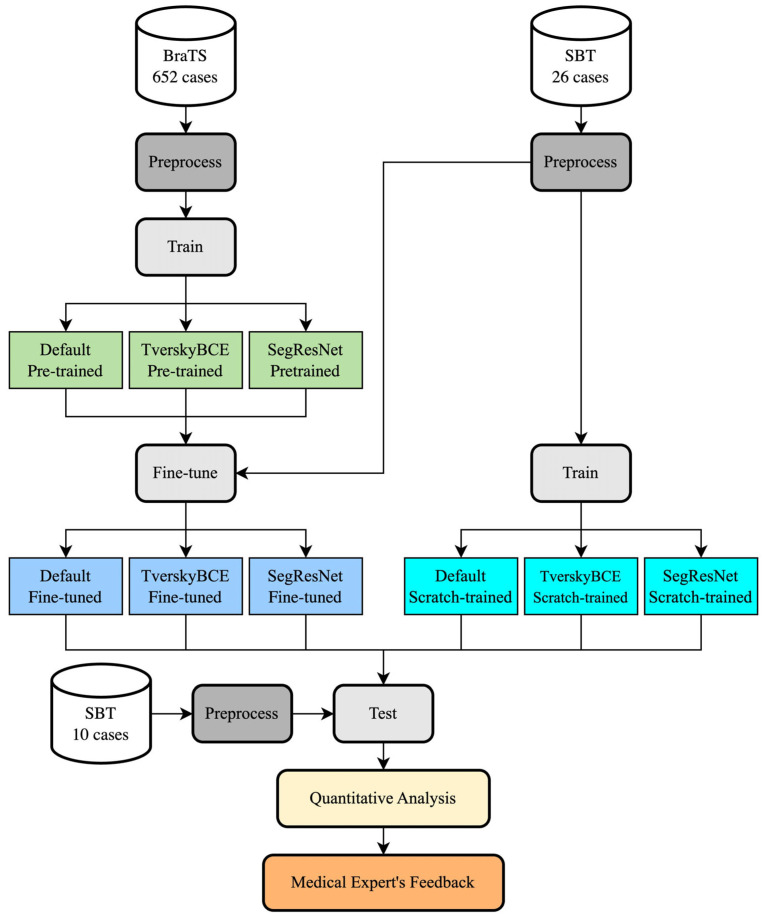
Overall process of training and fine-tuning models for brain metastasis segmentation.

**Figure 2 biomedicines-12-02561-f002:**
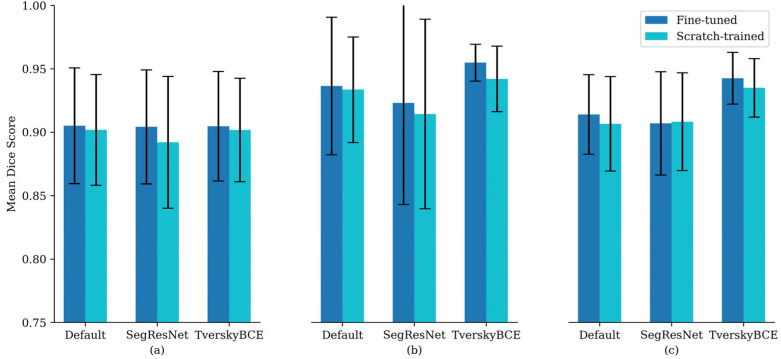
Comparison of mean Dice scores between fine-tuned and scratch-trained models on three brain tumor sub-regions: (**a**) enhancing tumor, (**b**) tumor core, and (**c**) whole tumor. The error bars indicate the 95% confidence interval.

**Figure 3 biomedicines-12-02561-f003:**
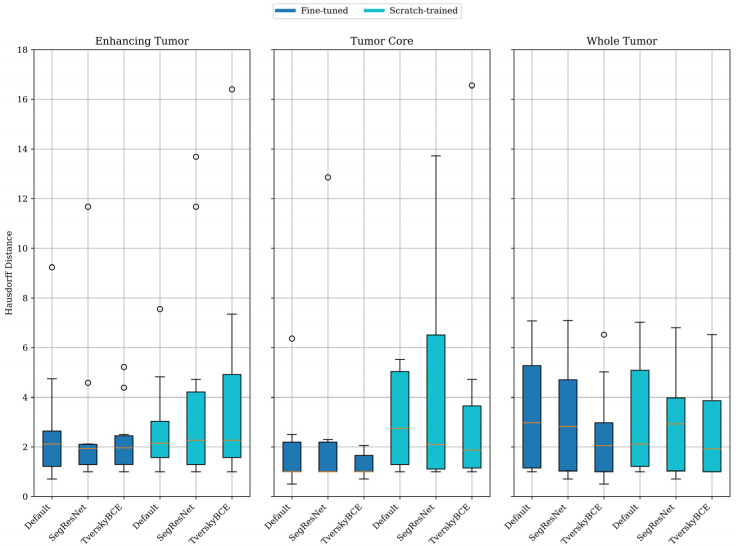
Comparison of mean Hausdorff Distances between fine-tuned and scratch-trained models on three brain tumor sub-regions. The circles indicate outliers (unusually high or low values), while the orange line within each box represents the median value, showing the central tendency of the Hausdorff Distances for each model.

**Figure 4 biomedicines-12-02561-f004:**
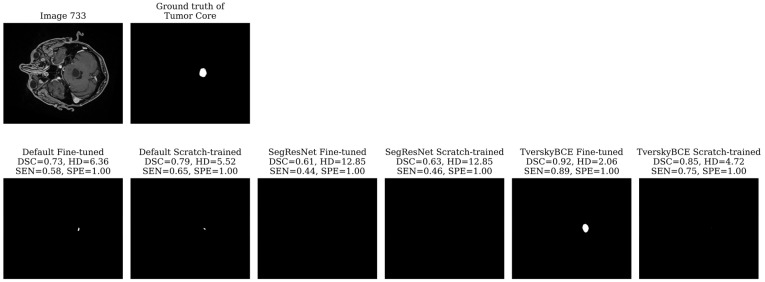
Segmentation results for test sample 733. TverskyBCE Fine-tuned achieved the highest DSC and lowest HD, successfully capturing the small tumor core missed by other models.

**Figure 5 biomedicines-12-02561-f005:**
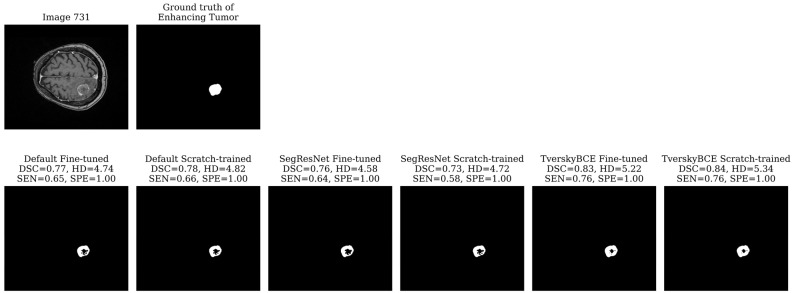
Segmentation results for Test sample 731. TverskyBCE Fine-tuned had the fewest missed regions for the enhancing tumor, with the highest SEN.

**Figure 6 biomedicines-12-02561-f006:**
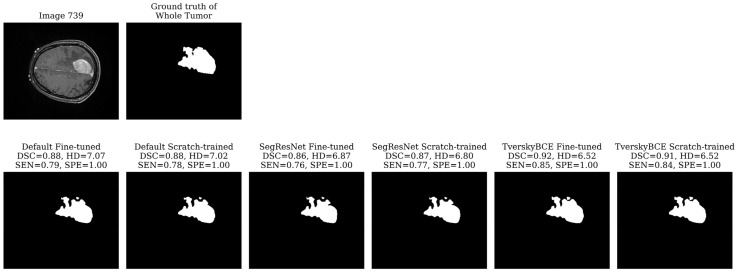
Segmentation results for Test sample 739. TverskyBCE Fine-tuned captured the large, irregular tumor more effectively, with the highest DSC and SEN.

**Figure 7 biomedicines-12-02561-f007:**
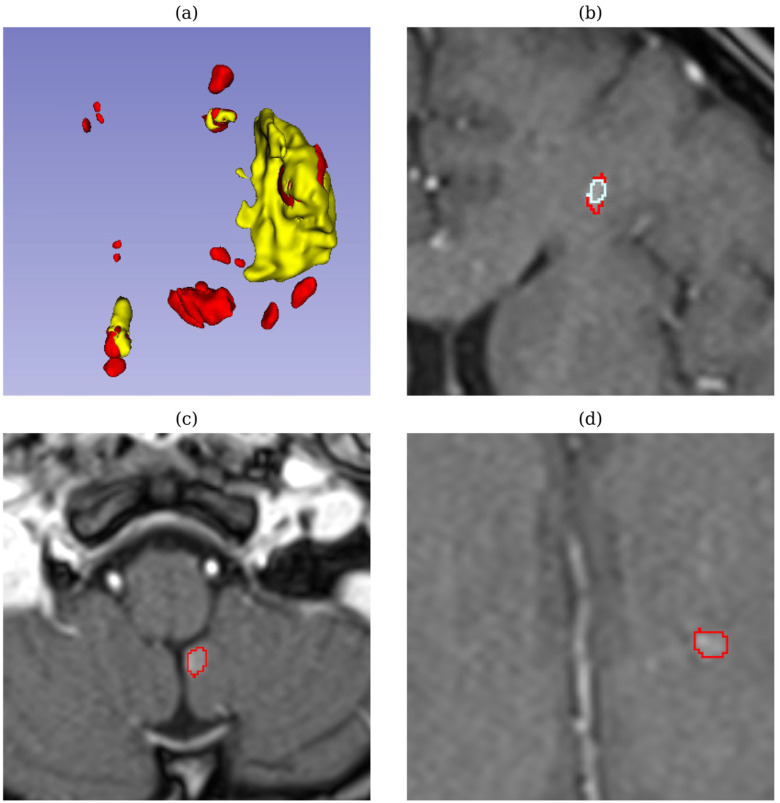
Segmentation results for case 728 (breast metastasis). (**a**) Ground truth 3D image with annotations marking multiple tiny, scattered tumors (yellow for peritumoral edema, red for the GD-enhancing tumor); (**b**) a 2D slice showing TverskyBCE segmentation, with red outlines for expert annotations and cyan for TverskyBCE’s detection of a small metastasis missed by SegResNet; (**c**,**d**) 2D slices showing tiny tumor regions missed by both models, with red outlines added by medical experts to mark these undetected areas.

**Figure 8 biomedicines-12-02561-f008:**
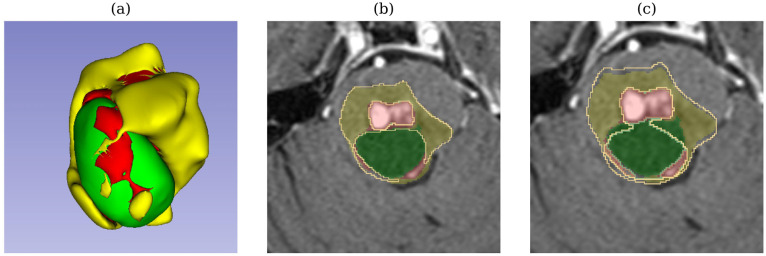
Visual comparison of segmentation results for test sample 729 (adenocarcinoma metastases) showing (**a**) the ground truth annotated by medical experts, (**b**) SegResNet segmentation where the necrotic tumor core (the green area) was missed, and (**c**) TverskyBCE segmentation successfully detecting the necrotic core. Solid-colored regions represent the ground truth annotations, while the outlined regions depict the model predictions. The annotations use yellow for peritumoral edema, red for the GD-enhancing tumor, and green for the necrotic tumor core.

## Data Availability

The BraTS Metastases 2024 dataset presented in this study is openly available at https://www.synapse.org/Synapse:syn59059764 (accessed on 15 July 2024). The Siberian Brain Tumor dataset presented in this article is not readily available due to privacy and ethical restrictions, as it contains private health data collected from our Federal Neurosurgical Center. Sharing it publicly would violate patient privacy and confidentiality agreements in accordance with ethical and regulatory standards. Requests for access to this dataset may be considered on a case-by-case basis, contingent upon appropriate ethical review and agreements.

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
