# Peer review of "Transfer Learning Approaches for Brain Metastases Screenings"

_biomedicines, 2024, doi:10.3390/biomedicines12112561_

Round 1

Reviewer 1 Report

Comments and Suggestions for Authors

The article describes the efficiency of transfer learning in automatic segmentation of brain metastases on MRI. First, three deep-learning models were trained. Results were then compared between pretrained models and models trained from scratch.

 The introduction, gives us a good insight into the current state of deep-learning in medical image analysis. The materials and method section describes the methods in detail, how two datasets were used and the three models were trained. Here, I would suggest that the authors summarize this section using a flow-chart to improve the overall understanding of the methods employed.

The results section gives is written correctly and difficult cases, described at the end of the section, underline the current difficulties with automatic analysis. The figures appropriately present various results. The discussion section summarizes and supports the results, describes key challenges and weaknesses of current learning models.

Overall, it is a novel and well performed study. After some minor corrections are made, the article is suitable for publishing in the journal.

Comments on the Quality of English Language

Minor editing needed

Author Response

Thank you very much for taking the time to review this manuscript. Please find the response below.

Comment: "summarize this section using a flow-chart to improve the overall understanding of the methods employed"
Response: We agree. The flowchart has been added to better summarize the section and improve overall comprehension of the methods used. Please see the attachment.

Reviewer 2 Report

Comments and Suggestions for Authors

The submission paper is well written. The context describes the data materials and the effectiveness of deep learning models for brain MRI screening analysis. The evaluation and validation of the comparative analysis results are reasonable. The usage of AI-assisted tool in the study is all right.

The following comments could be considered to improve the manuscript:

1) The numbers of MRI scans and the metastasis cases of the ASNR-MICCAI data set should be described in Section 2.2.

2) In Section 2.3, architectures of the TverskyBCE and SegResNet could be plotted with figures for better illustration.

3) In Figure 6(a), the multiple tiny and scattered tumors should be marked with annotations in this image.

4) It is better to clarify the sample ID assignment. For example, the SBT data set contains 496 patients (36 metastasis cases) and 10 of 36 cases were used for testing. In Section 3.2, Test samples 733, 731, and 739 were from the testing cases. It is a little bit confused that if the 733 is the subject ID or assigned with new number in the whole (ASNR-MICCAI + SBT) data set.

Author Response

Thank you very much for taking the time to review this manuscript. Your valuable comments have helped us improve our manuscript. Please find the detail responses below.

Comment 1: The numbers of MRI scans and the metastasis cases of the ASNR-MICCAI data set should be described in Section 2.2

Response 1: We added number of MRI scans and metastasis case in Section 2.2, line 143.

Comment 2: In Section 2.3, architectures of the TverskyBCE and SegResNet could be plotted with figures for better illustration

Response 2: We agree with you and we have provided detailed plots of all network architectures and configurations in the Supplementary Materials. Given the large image sizes and the fact that this study did not introduce a new architecture, we chose to include them as supplementary materials to avoid disrupting the reading flow in the main manuscript. Interested readers can download these images for an in-depth review.

Comment 3: In Figure 6(a), the multiple tiny and scattered tumors should be marked with annotations in this image.

Response 3: In Figure 6(a) (now Figure 7(a)), we confirm that the image does indeed display the annotated ground truth of tiny, scattered tumors. To offer a comprehensive view of the entire tumor distribution, we presented it in 3D rather than as a 2D slice. Additionally, we have updated the figure caption to clarify this and address your concern.

Comment 4: It is better to clarify the sample ID assignment. For example, the SBT data set contains 496 patients (36 metastasis cases) and 10 of 36 cases were used for testing. In Section 3.2, Test samples 733, 731, and 739 were from the testing cases. It is a little bit confused that if the 733 is the subject ID or assigned with new number in the whole (ASNR-MICCAI + SBT) data set.

Response 4: We agree with you and we added explanation of test sample IDs in section 2.2, line 160-163.

Round 2

Reviewer 2 Report

Comments and Suggestions for Authors

The revised manuscript is satisfied now.